# Laboratory Evaluation of Pupal Parasitoids for Control of the Cornsilk Fly Species, *Chaetopsis massyla* and *Euxesta eluta*

**DOI:** 10.3390/insects13110990

**Published:** 2022-10-28

**Authors:** Sandra A. Allan, Christopher J. Geden, J. Lanette Sobel

**Affiliations:** 1Center for Medical, Agricultural and Veterinary Entomology, Agricultural Research Service, United States Department of Agriculture, Gainesville, FL 32608, USA; 2Entomology and Nematology Department, University of Florida, Gainesville, FL 32608, USA

**Keywords:** biological control, *Muscidifurax*, *Nasonia*, *Spalangia*, sweet corn, Ulidiidae, management

## Abstract

**Simple Summary:**

Cornsilk flies are highly destructive flies that cause damage to developing cobs of corn, particularly sweet corn. Eggs laid on cornsilk hatch and larval feeding on the kernels under the protective outer leaves results in damage and often render cobs unsaleable. Mature larvae leave the cobs and pupate outside of the cob. The aim of this study was to evaluate the potential of known fly pupal parasitoids for use as biological control agent for these flies. Five species of commercially available parasitoids were evaluated, and all developed in cornsilk fly pupae and caused significant mortality. Our results indicated that these parasitoid species warrant further consideration for their potential (mortality range 46.1–98.5%) against cornsilk flies in the field.

**Abstract:**

Cornsilk flies are serious pests of sweet corn through damage to cobs and secondary fungal establishment. As pupae are generally outside the infested cob on the ground, there can be potential for use of pupal parasitoids for control. Two species of gregarious parasitoids, *Muscidifurax raptorellus* and *Nasonia vitripennis*, and three species of solitary parasitoids, *Spalangia endius*, *Spalangia cameroni* and *Muscidifurax raptor*, were evaluated against pupae of the two cornsilk fly species, *Euxesta eluta* and *Chaetopsis massyla.* House fly pupae, the most common host for most of the parasitoids, were included for comparison. All of the parasitoids killed and successfully parasitized pupae of the two cornsilk fly species at rates that were similar to house fly pupae. Adult parasitoids that emerged from cornsilk fly hosts were somewhat smaller than parasitoids reared from house flies and had proportionally fewer females. These parasitoids, which are widely and commercially available for filth fly control, warrant further consideration for their potential against cornsilk flies in the field.

## 1. Introduction

Sweet corn (*Zea mays* L.) production in tropical and semitropical climates is threatened by several species of cornsilk flies that specifically target developing cobs for larval production. In Florida alone, fresh market sweet corn production can undergo heavy losses, with untreated fields reporting up to 100% loss [1,2,3]. Oviposition by females on the cornsilk of developing cobs results in damage from larval feeding on silk and kernels along with secondary fungal contamination, resulting in dramatically decreased levels of marketable corn [4,5,6]. Fly species contributing the most to damage and crop loss in Florida include *Euxesta eluta* Loew, *Euxesta stigmatias* Loew and *Chaetopsis massyla* Walker [6,7,8,9]. Varieties of sweet corn including a BT11 gene are not protected from feeding by corn silk flies necessitating frequent insecticide applications to protect the cobs [10]. Additionally, resistance to commonly used pyrethroids have been reported further complicating control efforts [11]. Pesticide applications are targeted against adults through direct contact or leaves. Control efforts are further complicated by the location of larvae, protected by husks while on corn ears, and pupae, present in the soil as both are largely protected from pesticide residues [2,3]. Early season infestations on maize generally originate from populations located on local host plants that are not subject to pesticide applications. Reduction of these source populations has the potential to dampen damage observed in crops.

Cornsilk flies, as members of the superfamily Tephritoidea, bear many similarities to tephritid fruit flies. Biological control of tephritid flies has been examined with a variety of different species of larval or pupal parasitoids [12,13] and proposed for use as biological control agents [12,14]. Laboratory and field studies have indicated that tephritoid flies are susceptible to several pteromalid pupal parasitoids that also commonly attack “filth flies” such as house flies and stable flies [15,16,17]. These parasitoids, mostly in the genera *Muscidifurax*, *Spalangia*, and *Nasonia*, have a wide host range and similar life histories [18]. Female parasitoids deposit venom and either a single egg (solitary species such as *Muscidifurax raptor*, *Muscidifurax zaraptor* and *Spalangia* spp.) or multiple eggs (gregarious species such as *Nasonia vitripennis* and *Muscidifurax raptorellus*) through the host puparium and onto the surface of the developing fly within. The eggs hatch and parasitoid larvae feed externally on the host. Pupation occurs within the host puparium and adults emerge by chewing and escaping through an emergence hole after ca. 14–28 days at 27 °C, depending on the species [18].

The objective of this study was to evaluate the potential for various fly pupal parasitoid species to aid in management of cornsilk fly species. Species of parasitoids readily colonizable and commercially available were evaluated and included both solitary and gregarious species.

## 2. Materials and Methods

### 2.1. Fly Rearing

Cornsilk flies, *Euxesta eluta* and *Chaetopsis massyla*, were reared in the laboratory based on material from Gregg Nuessly (University of Florida). They were provided 10% sugar water solution on cotton balls and provided with diet cups containing a corn/agar diet based on Owens et al. [19] supplemented with fresh green pepper sections throughout development. Diet was placed in plastic souffle cups (118.3 mL, Dart Container Corporation, Mason, MI, USA) for *E. eluta* (late-stage instars crawl or jump out of cups prior to pupation [2,20] and in waxed cardboard cartons (236 mL, Choice Paper Company, Brooklyn, NY, USA) for *C. massyla* (larvae burrow into cardboard layers prior to pupation). Diet cups were placed on sand or layers of 2-ply paper towels in silicon-gasketed plastic boxes (LocknLock, Seoul, Korea) with screened inserts in the lids. Pupae for use in assays were collected from the sand by sieve, from paper towel layers, or cardboard layers of diet cups. Flies were maintained at 27 °C and 60–80% RH under a 16:8 L:D photoperiod. House fly pupae for parasitoid rearing collected following standard procedures [21]. To determine average weight of pupae, samples of 30 pupae were obtained for two rearing cohorts of each species and weighed.

### 2.2. Parasitoid Rearing

Colonies of *M. raptor*, *M. raptorellus*, *Spalangia endius* and *Spalangia cameroni* were reared on house fly pupae [22,23]. Adults were maintained on house fly pupae (2 day old) at a ratio of ~10:1 (fly: parasitoid) that were provided 2x weekly. A colony of *Nasonia vitripennis* was initiated from adults obtained commercially (Arbico Organics, Oro Valley, AZ, USA) that were verified to species and similarly raised on house fly pupae. Parasitoids were maintained at 27 °C and 60–80% RH reared under constant darkness or low light conditions [23].

### 2.3. Bioassay Protocol

Laboratory assays were conducted with combinations of parasitoid species and cornsilk fly species with house flies included as controls. A group of 5 female parasitoids (3–5 days old) were placed in a screen-topped cup (59.1 mL, P200N, Dart, Mason MI, USA) containing either 20 pupae for ulidiid pupae or 50 pupae for *Musca* pupae for 24 h, then parasitoids were removed from the cups and pupae held for emergence. All pupae used in assays were 2–4 days old. Assays were conducted in 3–5 cohorts with unparasitized host controls as well as fly/parasitoid treatments. A total of 30–40 cups were evaluated for each fly/parasitoid combination. Bioassay cups were held at 27 °C and 60–80% RH under a 16:8 L:D photoperiod. After parasitoid emergence, data collected for each cup included numbers of: emerged flies, emerged male and female parasitoid progeny, pupae with parasitoid exit holes and pupae with no exit holes (not developed, uneclosed pupae). Calculations were made of the percentages of pupae that were killed as well as those successfully parasitized. number of pupae successfully parasitized as well as the number of parasitoids produced per pupa. Additionally, the number of parasitoids produced per host species and the percentage of those parasitoids that were female were calculated. For gregarious parasitoid species, the number of adult parasitoid progeny produced per parasitized pupa was also calculated. Residual mortality was calculated as the percent of dead pupae that produced neither a fly nor a parasitoid. This mortality, sometimes referred to as “dudding”, reflects the sum of innate host mortality plus mortality inflicted by parasitoids due to host-feeding, envenomization without oviposition, and failed development of parasitoid immatures [24]. If mortality of the corresponding unparasitized host controls in each group was >15%, then results were discarded and the trial repeated.

### 2.4. Morphometric Comparison

Tibial measurements of parasitoids are considered a reflection of overall parasitoid size [25,26,27]. For each parasitoid/host fly species combination, the right hind tibia from females was removed from 15 parasitoids. Each tibia was mounted in 10% bovine serum albumen on a glass slide and digital images and measurements obtained using a Keyence microscope (VH-S30K) (Keyence, Osaka, Japan) with a high-performance zoom lens (VH-Z100R).

### 2.5. Data Analyses

Comparisons of pupal weights between the two rearing cohorts were compared for each species using paired *t*-tests (*p* < 0.05). After normality testing (Shapiro–Wilk test), comparisons between pupal weights for each species were tested for Kruskal–Wallis ANOVA on ranks and comparison between means were made using Dunn’s Method. For each parasitoid species, comparisons of tibia length of females reared from each fly host species were made by ANOVA. Data were tested for normality (Shapiro–Wilk test) and if data were not normal, Kruskal–Wallis ANOVA on ranks was conducted with means comparisons using Dunn’s Method.

Attributes of host mortality and parasitism were compared for each parasitoid species between host fly species used. For each comparison, data were tested for normality (Shapiro–Wilk test), then data analyzed either with ANOVA followed by means comparisons using Student Neuman Keul’s test, or if not normal, Kruskal–Wallis ANOVA on ranks and comparison between means were made using Dunn’s Method. Analyses were conducted using SigmaStat v. 4.0 (Systat Software, San Jose, CA, USA).

## 3. Results

Pupal weights for *M. domestica*, *C. massyla* and *E. eluta* were 21.45 (±0.31) mg, 4.24 (±1.42) mg and 3.92 (±0.97) mg, respectively. There were no differences between the two rearing cohorts of pupae for each species (*p* > 0.05) and data were combined. There were significant differences between fly species (H = 120.12, df = 2, *p* < 0.0001) with house fly pupae heavier than the two other species and no difference between the two cornsilk fly species. Mean percent control mortality of fly pupae was similar for *M. domestica* (5.8 ± 0.7) and *E. eluta* (6.0 ± 1.0), with somewhat higher control pupal mortality for *C. massyla* (12.1 ± 1.6).

All solitary parasitoid species were able to develop and produced adults in all of the host fly species evaluated (Table 1). Parasitism by *M. raptor* resulted in high levels of host mortality (91.2–98.0%) with no difference between fly host species. Levels of successful parasitism differed between species, with highest parasitism in *C. massyla* (76.9%), followed by *M. domestica* (60.4%) and *E. eluta* (49.0%). Residual mortality was lower for *C. massyla* (21.8%) than for the other two host species, which did not differ from each other (21.8%, 33.5%). Significantly more parasitoid progeny emerged from *E. eluta* (0.71) and *C. massyla* (0.71) pupae than from *M. domestica* (0.45), however, a higher percentage of females was observed from *M. domestica* (64.5%) compared to the two cornsilk fly species (45.9%, 52.8%).

For *S. cameroni*, highest mortality was observed with *C. massyla* (82.7%) followed by *M. domestica* (65.6%) with lowest mortality with *E. eluta* (46.1%) (Table 1). Levels of pupal parasitism did not differ between species and ranged between 34.8–36.1%. Residual mortality was highest in *C. massyla* (60.7%), followed by *M. domestica* (55.6%), and *E. eluta* (24.5%).The number of progeny of parasitoids from host fly species was similar between all species (0.29–0.32). Of these, the highest percentage of parasitoids that were female was greatest in *M. domestica* (58.9%), with lower levels from *C. massyla* and *E. eluta* (39.9–46.4%).

For *S. endius*, mortality was highest in *C. massyla* (77.6%), followed by *E. eluta* and *M. domestica* (53.4%) (Table 1). Pupal parasitism was highest with *C. massyla* (42.9%) and low for *E. eluta* and *M. domestica* (12.4–17.3%). Residual mortality was highest in *M. domestica* (85.4%), followed by the two cornsilk flies, which did not differ from each other (52.3–56.8%) The number of parasitoids produced per pupae was greatest for *C. massyla* (0.4) and lower for *E. eluta* and *M. domestica* (0.12). There was no difference between species in terms of percentage of females produced (53.2–62.2%). Sex ratios of *S. endius* did not differ among parasitoids that emerged from the three host species.

For the gregarious *M. raptorellus*, mortality was highest with *C. massyla* pupae (98.5%) with slightly but significantly lower mortality with *E. eluta* and *M. domestica* pupae (86.6–87.5%) (Table 2). Pupal parasitism was greatest for *E. eluta* (76.6%) followed by *C. massyla* and *M. domestica* (45.2–54.1%). Residual mortality was highest among *C. massyla* exposed to *M. raptorellus* (54.1%), followed by *M. domestica* (38.7%) and *E. eluta* (14.7%). The number of adult parasitoids that emerged per parasitized pupa was higher from *M. domestica* and *C. massyla* (2.99 and 2.87, respectively) than from *E. eluta* (1.90). A higher percentage of adult progeny that were female were observed from *M. domestica* (52.2%) followed by lower percentages from *E. eluta* and *M. domestica* (30.8–32.6%).

For *N. vitripennis*, there was no difference in the percentage of host pupae that were killed (50.1–61.0%) among fly species (Table 2). Similarly, there was no difference among host species in terms of successful parasitism (24.7–28.5%). Residual mortality was higher in *C. massyla* (66.7%) than in *E. eluta* (43.2%) but both similar to *M. domestica* (59.1). There was greatest production of parasitoids per host pupae from *M. domestica* (0.88) with lower production from *E. eluta* and *C. massyla* (0.27–0.42). The number of adult parasitoids that emerged per parasitized pupa was highest from *M. domestica* (5.63) followed by *C. massyla* (2.18) and *E. eluta* (1.07).Of the parasitoids that emerged, the highest percentage that were female were from *M. domestica* (78.9%), followed by *C. massyla* (66.8%) and *E. eluta* (56.7%).

There were significant differences in tibia lengths of parasitoids reared from both solitary and gregarious parasitoid species, with the larger *M. domestica* hosts generally producing parasitoids with longer tibia (Table 3). The exception was *S. cameroni*, whose tibia did not differ in size among host fly species. For solitary species, tibia of *M. raptor* were largest from *M. domestica*, followed by *C. massyla* and *E. eluta*, whereas tibia of *S. endius* reared from *M. domestica* were larger than those from *E. eluta* and *C. massyla*, which were similar in size. For both of the gregarious parasitoid species, tibia were largest among parasitoids reared from *M. domestica*, followed by *C. massyla* and *E. eluta*.

## 4. Discussion

Chalcidoid parasitoids of filth flies have broad host ranges that include Diptera of many families, including members of the Tephritoidea. *Spalangia endius*, which has long been used for house fly management [28], has also been found to parasitize at least 13 species of tephritids, including economically important species of *Anastrepha*, *Ceratitis*, and *Dacus* [29]. *Spalangia cameroni*, also known primarily as a muscid fly parasitoid, also parasitizes *A. suspensa* and two *Dacus* species [30]. *Pachycrepoideus vindemmiae* is most commonly collected from filth fly hosts but has also been collected from medfly pupae in several locations [31,32,33].

Because these parasitoids are easily and economically reared on wide range of hosts, there has been considerable interest in evaluating them for use as augmentative biological control agents against tephrititoid pest pupae. *Spalangia cameroni* has been studied extensively, with mass-rearing methods developed to produce parasitoids on medfly hosts [34,35,36,37,38,39], host preferences for medfly and house fly [40], and interactions with other biological control agents of medfly [41]. Development of *Spalangia endius* was studied in guava fruit fly (*Bactrocera correcta*) and oriental fruit fly (*B. dorsalis*) [17,42], and a mass-rearing system was developed using medfly hosts [43]. Although there are no known reports of *Muscidifurax* spp. attacking tephrititoid flies, *M. raptorellus*-medfly interactions have been studied under laboratory conditions [44,45], and *M. raptor* was used successfully against medflies in a South African vineyard [16]. All of these parasitoids are available from commercial insectaries in the United States and in many other countries.

In the present study, all of the tested parasitoids killed and developed successfully in two cornsilk fly species that are much smaller than their typical house fly hosts. These results are consistent with many of the above studies and with those of Geden and Moon [22] and Geden et al. [23], who observed successful development of *Spalangia* and *Muscidifurax* spp. in host sizes ranging from 5.5 [horn fly, *Haematobia irritans*] to >100 mg (*Sarcophaga bullata*). Although *N. vitripennis* has a wide host range, Rivers and Denlinger [46] found that this species did not successfully parasitize pupae of the tephritid *Rhagoletis pomonella* [Walsh] (apple maggot fly). Although, in this study house fly pupae were over five times larger than cornsilk flies, the parasitoids successfully parasitized the smaller cornsilk flies at rates equal to or higher than house fly hosts. Moreover, adult parasitoids that emerged from cornsilk flies were only marginally smaller than those reared on house flies.

*Muscidifurax raptorellus* produced about the same number of adult progeny per parasitized host pupa on *C. massyla* as on house fly hosts, indicating that this species does not take host size into account when deciding how many eggs to deposit. In another study, this species produced about the same number of adult progeny per parasitized host (2.2–2.9) on horn fly hosts as on *Sarcophaga bullata*, which were about 20 times larger. In contrast, *N. vitripennis* produced 2–4 times as many progeny per host on house flies than on the cornsilk flies. Although *N. vitripennis* sometimes attacks house flies, it is best known as a parasitoid of calliphorids and sarcophagids and is a model system for studying parasitoid behavior, sex-determination, physiology and genetics [47,48]. Clutch size has been studied extensive in this species, and it is well-known for making allowances of clutch size based on host size and quality [46,49,50]. Our results provide further support for the plasticity of *N. vitripennis* in allocating eggs based on host size.

All of the parasitoids except *S. endius* produced proportionally more females on house fly hosts that on the smaller cornsilk flies. Sex allocation has been studied extensively in parasitic Hymenoptera because their haplodiploid sex determination allows for sex allocation adjustments depending on host quality, size, prior parasitization, and other factors [51,52]. Our results are consistent with general trend of parasitoids allocating greater proportions of female progeny in larger or higher quality hosts [53,54].

In the United States the only parasitoids reported from cornsilk flies have been *P. vindemmiae*, collected from laboratory colonies of *E. eluta* and *E. stigmatias* in south Florida [55]. This solitary primary and secondary parasitoid has a broad host range in cyclorrhaghous Diptera and considered a larval/pupal parasitoid [56,57]. In laboratory studies, parasitoid development was supported in both fly species [55]. As cornsilk fly pupae in the field are sequestered, they are notoriously difficult to locate and collect [55] and this may account for the limited data on field parasitism data. In central and south America, parasitism of *E. euxesta* and *E. mazorca* Steyskal has been reported with a *Spalangia* spp. (pupal parasitoid), *Dettmeria euxestae* Borgmeirer (Eucoilidae) (larval parasitoid) and *Euxestophaga argentinensis* Gallardo (Figitidae) (larval parasitoids) [58,59]. Little is known about parasitism of *C. massyla*.

## 5. Conclusions

In conclusion, several species of pupal parasitoids that are widely attacked and developed successfully on the cornsilk flies *Chaetopsis massyla* and *Euxesta eluta*. Further research is needed on the ability of parasitoids to locate these potential target hosts under more natural conditions such as buried in soil [60]. For field applications, parasitized pupae would need to be protected from the elements and predation. In addition, the number or release stations per unit area of corn field would need to be determined, as dispersal distances are known to be relatively short for *S. cameroni* [61], *M. raptorellus* [62], and *M. raptor* [63]. Use of parasitoids in environments with heavy insecticide use are challenging but potential exists for fields relying on BT corn to provide targeted fly control, areas with low insecticide susceptibility, or in areas of fly refuge such as non-host plants or abandoned non-treated fields [6].

## Figures and Tables

**Table 1 insects-13-00990-t001:** Effects of parasitism (means ± SE) by solitary pupal parasitoids on cornsilk flies (*C. massyla*, *E. eluta*) and house flies (*M. domestica*).

Parasitoid Species	Host Species	Host Fly Pupae			Parasitoids		N
		% Mortality	% Successfully Parasitized	% Residual Mortality	# Progeny/Pupa	% Female	
*M. raptor*	*C. massyla*	98.3 (0.6) ^a^	76.9 (2.5) ^a^	21.8 (2.4) ^a^	0.71 (0.04) ^a^	52.8 (3.9) ^a^	30
	*E. eluta*	91.2 (3.4) ^a^	49.0 (5.7) ^c^	44.9 (5.8) ^b^	0.71 (0.04) ^a^	45.9 (3.5) ^a^	34
	*M. domestica*	94.1 (2.3) ^a^	62.4 (3.1) ^b^	33.5 (2.8) ^b^	0.45 (0.03) ^b^	64.5 (3.3) ^b^	33
	*F*	1.93	12.59	7.66	14.10	4.90	
	*P*	0.1509	<0.0001	0.0008	<0.0001	0.0095	
*S. cameroni*	*C. massyla*	82.7 (2.6) ^a^	34.8 (5.2) ^a^	60.7 (5.5) ^c^	0.32 (0.05) ^a^	46.4 (0.2) ^b^	30
	*E. eluta*	46.1 (5.4) ^c^	36.1 (5.2) ^a^	24.5 (5.0) ^a^	0.29 (0.06) ^a^	39.9 (8.2) ^b^	30
	*M. domestica*	65.6 (4.6) ^b^	37.2 (5.4) ^a^	55.6 (4.8) ^b^	0.32 (0.04) ^a^	58.9 (3.8) ^a^	34
	*F*	14.92	0.13	14.71	0.08	3.91	
	*P*	<0.0001	0.877	<0.0001	0.923	<0.0001	
*S. endius*	*C. massyla*	77.6 (5.4) ^a^	42.9 (6.6) ^a^	52.3 (6.6) ^a^	0.40 (0.06) ^a^	53.2 (4.8) ^a^	22
	*E. eluta*	53.4 (5.8) ^b^	17.3 (2.4) ^b^	56.8 (6.7) ^a^	0.12 (0.03) ^b^	54.6 (8.0) ^a^	30
	*M. domestica*	53.4 (5.4) ^b^	12.4 (5.3) ^b^	85.4 (5.7) ^b^	0.12 (0.05) ^b^	62.2 (8.2) ^a^	21
	*F*	5.26	11.38	6.05	10.27	0.32	
	*P*	0.007	0.0017	0.0038	<0.0001	0.724	

Within each column for each parasitoid species, results from one-way ANOVA (*F* and *p* values) are presented. Furthermore, means followed by different letters are significantly different (*p* < 0.05) by multiple comparison tests.

**Table 2 insects-13-00990-t002:** Effects of parasitism (means ± SE) by gregarious pupal parasitoids, *M. raptorellus* and *N. vitripennis* on cornsilk flies, *C. massyla and E. eluta* and house flies (*M. domestica*).

Parasitoid	Host	Host Fly			Parasitoid			N
		% Mortality	% Successfully Parasitized	% Residual Mortality	# Progeny/Pupae	# Progeny Per Parasitized Pupa	% Female	
*M. raptorellus*	*C. massyla*	98.5 (1.5) ^a^	45.2 (2.5) ^b^	54.1 (0.1) ^c^	1.29 (0.09) ^a^	2.87 (0.2) ^a^	32.6 (3.1) ^b^	30
	*E. eluta*	87.5 (4.0) ^b^	76.6 (4.4) ^a^	14.7 (2.7) ^a^	1.41 (0.16) ^a^	1.90 (0.2) ^b^	30.8 (3.9) ^b^	30
	*M. domestica*	86.6 (3.4) ^b^	54.1 (3.4) ^b^	38.7 (2.6) ^b^	1.67 (0.14) ^a^	2.99 (0.1) ^a^	52.2 (2.7) ^a^	30
	*F*	3.90	14.68	60.50	1.99	9.60	12.85	
	*P*	0.0238	<0.0001	<0.0001	0.1425	<0.0001	<0.0001	
*N. vitripennis*	*C. massyla*	61.0 (3.9) ^a^	24.7 (4.0) ^a^	66.7 (4.8) ^a^	0.42 (0.08) ^a^	2.18 (0.45) ^b^	66.8 (5.0) ^ab^	43
	*E. eluta*	50.1 (7.4) ^a^	28.5 (6.3) ^a^	42.2 (7.3) ^b^	0.27 (0.06) ^a^	1.07 (0.17) ^b^	56.7 (7.2) ^b^	30
	*M. domestica*	56.8 (5.4) ^a^	24.8 (3.6) ^a^	59.1 (3.1) ^ab^	0.88 (0.00) ^b^	5.63 (0.77) ^a^	78.9 (3.3) ^a^	40
	*F*	0.96	0.21	5.53	52.76	15.50	4.72	
	*P*	0.3853	0.8106	0.0051	<0.0001	<0.0001	0.0113	

Within each column for each parasitoid species, results from one-way ANOVA (*F* and *p* values) are presented. Furthermore, means followed by different letters are significantly different (*p* < 0.05).

**Table 3 insects-13-00990-t003:** Length (mm) (SE) of the front right tibia of parasitoids reared from host pupae of different fly species.

Parasitoid	Parasitoid Species	Fly Host Species			F (H)	df	*p*
		*E. eluta* {n}	*C. massyla* {n}	*M. domestica* {n}			
coriSolitary	*M. raptor*	550.41 (2.77) ^b^ {16}	565.81 (4.60) ^c^ {15}	613.48 (2.61) ^a^ {16}	95.617	2, 46	<0.001
	*S. cameroni*	479.79 (2.94) ^a^ {15}	478.23 (3.51) ^a^ {16}	473.21 (3.58) ^a^ {15}	1.027	2, 45	0.367
	*S. endius*	438.57 (9.67) ^b^ {17}	458.38 (3.63) ^b^ {15}	492.05 (7.58) ^a^ {25}	(7.077)	2	<0.001
Gregarious	*M. raptorellus*	474.56 (6.00) ^b^ {33}	511.05 (8.53) ^ab^ {18}	568.47 (5.81) ^a^ {17}	(7.087)	2	0.029
	*N. vitripennis*	603.67 (2.30) ^b^ {15}	613.33 (4.95) ^ab^ {15}	628.76 (5.39) {15}	5.870	2, 47	0.005

Within each row, means followed by different letters are significantly different (*p* < 0.05). Statistics presented for ANOVA are F (parametric) or H (non-parametric).

## Data Availability

Data presented in this study are available upon request.

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
