# Peer review of "Laboratory Evaluation of Pupal Parasitoids for Control of the Cornsilk Fly Species, Chaetopsis massyla and Euxesta eluta"

_insects, 2022, doi:10.3390/insects13110990_

Round 1
Reviewer 1 Report
Overall, this study highlights the importance of several parasitoids could be used towards augmentative biocontrol control of the cornsilk flies. The simple summary, abstract, objectives, methodology, statistical analyses, results, and discussions are well developed and easy to understand.

Author Response
We appreciate your comments and thank you for your reviewing effort. I am glad to hear that it is easy to understand and read.
Reviewer 2 Report
The reviewed manuscript provides valuable evidence on the effectiveness of five commercially available pupal parasitoid species as biological control agents of two pest species commonly called "cornsilk flies", Euxesta eluta and Chaetopsis massyla, which cause severe damage to developing corn cobs, particularly sweet corn. Parasitoid species tested were two gregarious species, Muscidifurax raptorellus and Nasonia vitripennis, and three solitary species, Spalangia endius, Spalangia cameroni and Muscidifurax raptor. Results of the study showed that the five parasitoid species successfully developed on the pupae of both cornsilk flies and caused high mortality of these pests.
The manuscript involves a brief but well-designed study to evaluate the ability of pupal parasitoid species to kill both corn pests. Although the parasitoid species involved in the tests are well known to be good biological control agents of species of Diptera Cyclorrhapha or Muscomorpha associated with cattle barns and poultry farms, their performance on cornsilk fly puparia was not known.
With regard to the structure of the manuscript, the objectives of the study are well stated and are coherent with the experimental design proposed; the results obtained respond to the objectives of the study. The data were appropriately analyzed. The manuscript is clearly written and the theoretical framework is well presented. Although I am not a native English speaker, both the writing style and the language are fine for me. This is a study that deserves to be published.
Author Response
Thank you for your reviewing effort and the positive comments. We are excited about the potential of these parasitoids for these particular species and will continue to further examine many interesting questions to further develop this IPM tool.
Reviewer 3 Report
In this work, authors tested the ability of five pupal parasitoids (3 solitary and 2 gregarious) to parasitize cornsilk fly species Euxesta eluta and Chaetopsis massyla, two pests of sweet corn. The authors used the house fly pupae as control and measured different parameters to compare the performance of the parasitoids. The manuscript is well written, presents an adequate introduction, describes the methodology well, analyzes the data correctly, and presents the results and discussion adequately. Only a few details need to be changed, which I describe below.
L60: Write the scientific name of the species M. raptor, M. zaraptor without abbreviation the first time it is mentioned in the text (from the introduction) and add the descriptor's name. Apply in all cases where a species is mentioned for the first time.
L87: It would be desirable if you add a photo of the 3 species of parasitoids together so that the reader has a clear and quick idea of the differences in size.
L103-110: difficult to read. Consider adding numbers or subsections to separate better the parameters measured
L159: (C. massyla, E. eluta)… it is missing the word “and”
L160: Delete one dot.
Table 1: Check the letters in the column “#progeny/pupa” with the parasitoid S. endius.
Tables 1 & 2: What do the numbers in parentheses mean?...SE?
L176: Date of residual mortality are not coincident with Table 1
L196: Letters to differentiate the treatments are missing in the column “#progeny per parasitized pupae”.
L187-188: Values of “% residual mortalty” are not agree with Table 2.
L202: Data of residual mortality of C. massyla (59.1%) is not agree with Table 2. Which is the good one?
L225: It is Ceratitis…not Ceratitus
-L227: “two Dacus” not 2 Daus
Author Response
We did not expand the introduction as we feel that felt that it was balanced as it. Note that in the written comments, it does indicate that the introduction is adequate.
For the specific issues listed:
L60. Scientific names were spelled out upon first use
L87. We do not feel that a picture is imperative for this publication and thus we have not included one.
Line 103-110 Good point. This section was reworded to make it more clear
Line 159 corrected
Line 160 corrected
Table 1 – correction made
Table 1&2 corrected so that bracket meaning was added to title
Line 176 corrected
Line 196 letters added
Line 187-188 Numbers were corrected
Line 225 typo corrected
Line 227 typo corrected
There was a manuscript with comments attached and these have been addressed line by line.
All changes made as suggested on the text except:
Addition of footnote 5 for Sobel was not made as there is no other email address
We thank the reviewer for the thorough review that does improve the manuscript.
Reviewer 4 Report
In this manuscript, the authors reported the effects of parasitism by pteromalid wasps on cornsilk flies and house flies. All of the parasitoids, including two gregarious species, Muscidifurax raptorellus and Nasonia vitripennis, and three solitary species, Spalangia endius, Spalangia cameroni and Muscidifurax raptor can kill and successfully parasitize pupae of the cornsilk fly species, Euxesta elutaand Chaetopsis massyla with similar parasitism rates to house fly pupae. Adult parasitoids that emerged from cornsilk fly hosts had smaller body sizes and fewer females compared with those reared on house flies. The results are meaningful, which contribute to further use of these parasitoids to control cornsilk flies in the field. Moreover, the content of this manuscript is well organized.
However, the research in this manuscript seems to lack scientific hypotheses. The research content is relatively simple, and the information provided by this manuscript seems to be limited. Generally speaking, most organisms exhibit phenotypes that are based on trade-offs among different characters, such as between body size and development time, or between fecundity and longevity. The authors only reported the effects of host flies on offspring body size of parasitoids. The research would be more comprehensive if more biological characters of parasitoids including functional response was evaluated.
Author Response
Thank you for your response and we agree that the results are meaningful in terms of the applied aspect of crop pest control. This was a very focused project looking at the specific question of whether these commercialized strains of parasitoids would potentially have any effect against these smaller fly species. I agree that this opens up many interesting and exciting ecological and biological questions and these are the focus of upcoming research.
Round 2
Reviewer 4 Report
I have no other suggestions or comments.